# *Bifidobacterium adolescentis* Is Effective in Relieving Type 2 Diabetes and May Be Related to Its Dominant Core Genome and Gut Microbiota Modulation Capacity

**DOI:** 10.3390/nu14122479

**Published:** 2022-06-15

**Authors:** Xin Qian, Qian Si, Guopeng Lin, Minmin Zhu, Jingyu Lu, Hao Zhang, Gang Wang, Wei Chen

**Affiliations:** 1State Key Laboratory of Food Science and Technology, Jiangnan University, Wuxi 214122, China; qian_x@hotmail.com (X.Q.); 6170112068@stu.jiangnan.edu.cn (Q.S.); 7200112079@stu.jiangnan.edu.cn (G.L.); ljy@jiangnan.edu.cn (J.L.); zhanghao61@jiangnan.edu.cn (H.Z.); chenwei66@jiangnan.edu.cn (W.C.); 2School of Food Science and Technology, Jiangnan University, Wuxi 214122, China; 3Department of Anesthesiology, Wuxi No. 2 People’s Hospital, Wuxi 214002, China; 4National Engineering Research Center for Functional Food, Jiangnan University, Wuxi 214122, China; 5(Yangzhou) Institute of Food Biotechnology, Jiangnan University, Yangzhou 225004, China

**Keywords:** *Bifidobacterium*, type 2 diabetes, gut microbiota, inflammation, short-chain fatty acids, genome

## Abstract

The prevalence of diabetes mellitus is increasing globally. Probiotics have been shown to be an effective intervention for diabetes. This study focused on the relieving effects and possible mechanisms of 16 strains of two dominant *Bifidobacterium* species (*B. bifidum* and *B. adolescentis*, which exist in the human gut at different life stages) on type 2 diabetes (T2D). The results indicated that more *B. adolescentis* strains appeared to be superior in alleviating T2D symptoms than *B. bifidum* strains. This effect was closely related to the ability of *B. adolescentis* to restore the homeostasis of the gut microbiota, increase the abundance of short-chain fatty acid-producing flora, and alleviate inflammation in mice with T2D. In addition, compared with *B. bifidum*, *B. adolescentis* had a higher number of core genes, and these genes were more evolutionarily stable, including unique environmental tolerance, carbon and nitrogen utilization genes, and a blood sugar regulation gene, *glgP*. This may be one of the reasons why *B. adolescentis* is more likely to colonize in the adult gut and show a superior ability to relieve T2D. This study provides insights into future studies aimed at investigating probiotics for the treatment of metabolic diseases.

## 1. Introduction

Diabetes mellitus (DM) is a metabolic disease characterized by hyperglycaemia. Most cases of DM are type 2 DM (T2D) [1]. The T2D mainly manifests as insulin resistance, which, in turn, leads to obvious hyperglycaemia [2]. According to the latest survey results of the International Diabetes Federation, there are 460 million adults globally with diabetes (as of 2019). Based on this trend, the number of people with diabetes is likely to reach 700 million by 2045 [3].

Drug therapy is the main treatment for T2D. Biguanides and glitazones are traditional insulin sensitizers, and drugs such as sulfonylureas and non-sulfonylureas are traditional insulin secretagogues. The direct injection of insulin is also one of the methods used to treat T2D. In recent years, new initiators of insulin secretion, dipeptidyl peptidase-4 inhibitors, and glucagon-like peptide-1 receptor agonists, have been used in the clinical treatment of T2D [4]. Drug treatment is effective, but it also has certain adverse effects, such as weight gain, gastrointestinal discomfort, liver damage, myasthenia, and sexual dysfunction. Therefore, there is an urgent need to find more effective and safer substances to treat T2D.

Previous studies have shown that disturbances of the gut microbiota are associated with a variety of metabolic diseases, such as diabetes, obesity, and nonalcoholic fatty liver disease [5]. Compared with healthy mice, mice with T2D show an increase in the Firmicutes/Bacteroides (F/B) ratio and a decreased abundance of certain functional bacteria (such as *Bifidobacterium*). This is accompanied by an increased abundance of opportunistic pathogens and some lipopolysaccharide-producing gram-negative bacteria. Other studies have shown that the human gut microbiome changes before diabetes develops. Zhang et al. [6] detected and analyzed the fecal bacteria of 121 subjects in the three stages of (1) normal glucose tolerance, (2) pre-T2D, and (3) T2D using a 16S rRNA gene high-throughput sequencing method. They found that the abundance of acid-producing bacteria, such as *Akkermansia mucinipila* ATCCBAA-835 and *Faecalibacterium prausnitzii* L2-6, was lower in the intestines of subjects with pre-T2D compared with healthy subjects, and they speculated that the presence of *Verrucomicrobiae* may be a sign of T2D. In addition, a decrease in the number of microbial communities in the intestines and a decrease in bacterial diversity, accompanied by an increased abundance of pathogenic bacteria and a decreased abundance of beneficial bacteria, may lead to low-grade chronic inflammation due to intestinal immune system activity [7], which further leads to insulin resistance.

*Bifidobacterium* is a genus of gram-positive bacteria and one of the most important flora in the human and animal intestinal environment. Studies have shown that the abundance of *Bifidobacterium* is significantly decreased in the intestines of T2D patients [8], and supplementation with *Bifidobacterium* (*B. bifidum* or *B. adolescentis*) alleviates gut microbiota disorders and lowers blood glucose concentration [9,10]. However, the two strains appear and dominate in the intestines at different stages of human growth [11,12]. Correspondingly, T2D mostly occurs in adults. Thus, we sought to determine whether these two *Bifidobacterium* species differ in their ability to alleviate the symptoms of T2D.

In this study, eight strains of *B. bifidum* and eight strains of *B. adolescentis* derived from different populations were selected to analyze the differences in their effects on T2D metabolic parameters and visceral pathology. We further detected the specific compositional changes in the gut microbiota and the accompanying changes in SCFAs content and inflammatory status to assess the mechanism by which *Bifidobacterium* alleviate symptoms in a commonly used T2D model mice induced by high-fat diet and streptozotocin (STZ). In addition, we also compared the core genomes to explore the factors responsible for the differences in the capacity of the two different *Bifidobacterium* species to alleviate T2D, thus providing a reference for the selection of probiotics that alleviate metabolic diseases.

## 2. Materials and Methods

### 2.1. Preparation of Bacterial Strains

All of the experimental strains were obtained from the Food Biotechnology Center of Jiangnan University (Table 1), and were cultured in modified DeMan-Rogosa-Sharpe medium supplemented with 0.05% L-cysteine hydrochloride for approximately 16 h at 37 °C. The strains were then continuously cultured three times with a 1% (*v*/*v*) inoculum. The collected cells (8000× *g*, 5 min, 4 °C) were washed twice with sterile normal saline and then resuspended to 2 × 10^8^ colony-forming units/mL for gavage.

### 2.2. Animal Experiments

Three-week-old male specific-pathogen-free grade C57BL/6J mice (Slack, Shanghai, China) were raised at the Animal Experiment Center of Jiangnan University under a 12-h light/12-h dark cycle at 23 °C ± 2 °C and 50% ± 10% relative humidity. After 1 week of a normal diet and water to allow adaptation, 114 mice were randomly divided into the following 19 groups (6 mice/group): control, model, metformin, and 16 *Bifidobacterium*-treated groups. The details of the design of the animal experiments are shown in (Figure 1a). Mice in each group were fed a high-fat diet (the control group was fed a low-fat diet) and were administered metformin or a *Bifidobacterium* suspension by gavage for 12 consecutive weeks. The feed and gavage formulations are detailed in Appendix A. On the first day of week 7, the mice, except those in the control group, were injected with STZ (Sigma, St. Louis, MO, USA). Mice in all groups were fasted for 12 h in advance, the Control group were intraperitoneally injected with 50 mmol/L citric acid-sodium citrate buffer (pH 4.5), the other groups were intraperitoneally injected with STZ dissolved in 50 mmol/L citric acid-sodium citrate buffer at the dose of 100 mg/kg·bw. Then, blood glucose levels were determined at 0, 0.5, 1.0, and 2.0 h. The area under curve (AUC) was calculated as follows: AUC = 0.25 × (G0 h + G0.5 h) + 0.25 × (G0.5 h + G1.0 h) + 0.5 × (G1.0 h + G2.0 h). All of the animal procedures were performed in accordance with the experimental animal management regulations of the Animal Welfare and Ethics Committee of Jiangnan University (JN. No20190615c1261115).

### 2.3. Sample Collection and Processing

One day before the mice were killed, fresh fecal samples were collected in sterile 1.5 mL Eppendorf tubes and kept on ice throughout the processing procedure. After fasting for 12 h, the mice were intraperitoneal injected with a 1% pentobarbital sodium solution at a dose of 5 mL/kg, and blood samples were collected. The blood samples were centrifuged at 800× *g* for 15 min and the upper serum layer was collected. Pancreatic tissue was rinsed with normal saline precooled to 4 °C and preserved in formalin. Livers were weighed and divided into two parts, which were stored in formalin or liquid nitrogen. All of the collected samples were immediately stored at −80 °C.

### 2.4. Biochemical Analysis

The insulin resistance index (HOMA-IR) was calculated using the following formula: HOMA-IR = fasting insulin concentration × FBG concentration/22.5. Blood glucose was measured with Roche glucometer and blood glucose test strips by means of tail blood collection with a blood collection needle. High-density lipoprotein cholesterol (HDL-C), low-density lipoprotein cholesterol (LDL-C), total cholesterol (TC), and triglyceride (TG) concentrations were detected using commercial assay kits (Nanjing Jiancheng Bioengineering Institute, Nanjing, China). The serum concentrations of insulin, interleukin-6 (IL-6), tumor necrosis factor alpha (TNF-α), and interferon gamma (IFN-γ) were measured using an enzyme-linked immunosorbent assay kit (Senbeijia Biological Technology, Nanjing, China).

### 2.5. Histopathological Analysis

Pancreas and liver tissues were fixed in 4% paraformaldehyde, embedded in paraffin, and stained with hematoxylin-eosin (HE). Stained pancreas and liver tissues were observed under a microscope with 400× and 200× magnification scale, respectively. All morphological lesions were rated on a scale of 1–5 by their severity according to the NTP Pathology guidelines. Tissue sections with no lesions were marked as 0. Pancreatic tissues were scored for exocrine and endocrine. The exocrine scores were mainly based on the morphology of pancreatic cells and whether there were vacuolar deformation and turbidity. The endocrine scores were mainly based on the number of islets, the shape of the islets, and whether the cells showed nuclear condensation. The final score was the sum of the exocrine and endocrine scores. The scores of liver tissue lesions were mainly based on the status of steatosis, including the size and density of fatty vacuoles in the liver tissue. The pancreatic tissue and liver tissue of each mouse were all scored in a blinded manner [13].

### 2.6. SCFA Analysis

Short-chain fatty acids (SCFAs) were analyzed in the mouse fecal samples using gas chromatography-mass spectrometry (1300-ISQ-GC-MS; Thermo Fisher, Waltham, MA, USA). The extraction of SCFAs from mouse fecal samples and their subsequent analysis were performed as previously described [14].

### 2.7. Gut Microbiota Profiling and Bioinformatical Analysis of Genome

Total DNA was extracted from the mouse fecal samples using a Fast DNA Spin Kit (MP Biomedicals, Santa Ana, CA, USA). The V4 region of the bacterial 16S rRNA gene was amplified by polymerase chain reaction and the amplified products were subjected to 2% agarose gel electrophoresis at 100 V for 30 min. The products were recovered and quantified using a DNA Gel Purification Miniprep Kit (Biomiga, Hangzhou, China) and 50 μL libraries were constructed based on equal mass concentrations. The libraries were then sequenced on a MiSeq sequencer (Illumina, San Diego, CA, USA) [15], and the sequencing data underwent specialized bioinformatics analysis, as described in Appendix A.

The accession numbers of the genomic data used in this study are shown in Appendix A, and the bioinformatical analysis of genome are detailed in Appendix A.

### 2.8. Statistical Analysis

Experimental data were processed using GraphPad Prism 8 (GraphPad, San Diego, CA, USA) and SPSS 22 (IBM, Armonk, NY, USA) software. Data are presented as the mean ± standard deviation. Differences between groups were analysed using least squares difference analysis with one-way analysis of variance, and *p* < 0.05 was considered significant.

## 3. Results

### 3.1. B. adolescentis and B. bifidum Show Differential Effectiveness in Regulating Glucose and Lipid Metabolism Disorders in T2D Mice

FBG and OGTT concentrations are the main diagnostic indicators of diabetes and are also the key indicators for measuring blood sugar status. One week after STZ injection, the OGTT of the T2D mice showed a rapid increase and then a slow decline, remaining high 2 h after gavage. Additionally, the area under the blood glucose concentration curve (AUC_glucose_) was significantly higher in the model group than in the control group (*p* < 0.05), indicating abnormal blood glucose metabolism and impaired glucose tolerance (Figure 1b–d and Appendix A). Supplementation with different *Bifidobacterium* strains showed different effects on high-fat-diet-and-STZ-induced increases in blood glucose concentration. Mice supplemented with *B. adolescentis* 8M4, 26M1, 34M4, or 2016 or *B. bifidum* 26M7 had lower FBG concentrations than mice without probiotic supplementation, and none met the criteria for diabetes. It is worth noting that most *B. adolescentis* also exhibited a strong ability to inhibit the abnormal increase in PBG concentration. *B. adolescentis* 26M1, 41M3 and 2016 significantly decreased the AUC_glucose_ value and delayed the occurrence of glucose intolerance. After 5 weeks of STZ injection, the diabetic condition of the mice became more severe (Figure 1e–g and Appendix A). However, with the development of diabetes mellitus, the ability of *B. bifidum* strains to decrease PBG concentration was weakened (except for strain 9M10). In addition, based on the HOMA-IR values, most *B. adolescentis* strains improved the insulin sensitivity of the mice to varying degrees, while only 9M10 and 26M7 in *B. bifidum* strains showed improvement potential (Figure 1h,i).

Dyslipidaemia is a common feature of T2D and a major cause of its complications. Serum TG, TC, HDL-C, and LDL-C concentrations are usually used to determine blood lipid status. In this study, compared with the mice in the control group, those in the model group had significantly higher TC and LDL-C concentrations (Appendix A), which indicated that the T2D mice had abnormal blood lipid metabolism. Supplementation with *B. adolescentis* 3M10 and *B. bifidum* 5M8 decreased the concentrations of TG and LDL-C, and supplementation with *B. bifidum* 21M3 decreased the concentration of LDL-C. In addition, other *Bifidobacterium* strains, such as *B. adolescentis* 2016 and *B. bifidum* 26M7, showed a tendency to decrease the serum LDL-C concentration of the T2D mice, but this effect was not significant.

### 3.2. B. adolescentis Strains Showed Superior Potential Than B. bifidum Strains in Attenuating Pancreatic and Liver Damage in T2D Mice

A high-fat diet combined with STZ impaired insulin sensitivity in diabetic mice, and some *Bifidobacterium* species alleviated this effect. Therefore, we further investigated whether the administration of the two *Bifidobacterium* species also had a positive effect on pancreatic morphology, as determined by HE staining. The islet cells of T2D mice showed altered morphology and decreased number, and the cells appeared pyknotic, but *Bifidobacterium* treatment significantly improved these pancreatic lesions (Figure 2a). According to the pathological score of pancreas tissue sections (Figure 2b), all strains of *B. adolescentis*, except for 30M5 and 50M3, improved the fatty lesions of mice pancreata to varying degrees. However, only 2 *B**. bifidum* strains were shown to prevent pancreatic damage.

In addition to pancreatic islet damage, the livers of diabetic model animals and clinical patients also have lesions. In this study, the weight of the mice with T2D and the control mice did not differ significantly, but the liver/body weight (%) significantly increased in the mice with T2D (Appendix A). The livers of the mice in the model group showed a large amount of fatty vacuole degeneration, but *B. adolescentis* treatment significantly improved these liver lesions (Figure 2c,d). However, the effect of *B. bifidum* on the liver fatty lesions in the mice with T2D was limited. In addition, *B. adolescentis* 8M4 and 2016 showed the greatest effect and also inhibited the abnormal liver/body weight (%) increase caused by T2D (Appendix A).

Besides, oxidative stress is closely related to the occurrence and development of diabetes and its complications, and there is a persistent vicious cycle between oxidative stress and disorders of glucose and lipid metabolism. The liver is the main site of oxidative damage in mice [16]. Malondialdehyde (MDA) is the final product of lipid peroxidation, and its concentration may reflect the degree of lipid peroxidation in the body. The activity of superoxide dismutase (SOD) reflects the ability of the body to scavenge oxygen free radicals. The hepatic MDA concentration was significantly higher in the mice with T2D than in the control mice, but the SOD activity was normal (Figure 2e,f). After the administration of *B. adolescentis*, hepatic MDA concentrations were decreased to varying degrees. Strain 2016 had the greatest effect, and this strain also significantly increased the hepatic SOD activity. While none of the *B. bifidum* strains, with the exception of strain 9M10, showed significant effects on oxidative stress.

Blood glucose-related indicators (blood glucose concentration, glucose tolerance, insulin resistance and pancreatic islet damage), blood lipid indicators, and organ fat deposition indicators were analyzed by principal component analysis, and the effects of the two bifidobacteria species on high-fat-diet-and-STZ-induced T2D were compared. *B. adolescentis* strains showed an advantage over *B. bifidum* strains in alleviating T2D symptoms, and some strains were more effective than metformin. None of the *B. bifidum* strains, except for 9M10 and 26M7, showed a significant remission effect (Figure 2g).

### 3.3. Bifidobacterium Strains Restore Gut Microbiota Homeostasis in Mice with T2D and Alleviate Symptoms

In view of the difference in the effects of the two *Bifidobacterium* species on the symptoms of mice with T2D, we sequenced the fecal microbiota of the mice to determine whether the observed difference was due to the modulation of the gut microbiota. A high-fat diet combined with STZ treatment resulted in a significant increase in the Shannon index and Simpson index values of the gut microbiota, and slight, but nonsignificant, decreases in the Chao1 index value and the number of observed species (Figure 3a–d). The administration of *B. adolescentis* strains, except for 26M1, increased the richness of the gut microbiota to varying degrees, and strains 26M1, 30M5, and 2016 inhibited the abnormal increase in Shannon and Simpson index values in the mice with T2D. While the administration of some *B. bifidum* strains resulted in a significant change in a single microbiota-related index value in the mice with T2D, the majority of *B. bifidum* strains showed no significant regulatory effect on the alpha diversity of gut microbiota. β-diversity analysis showed that a high-fat diet combined with STZ treatment led to significant changes in the microbiota structure of the mice (Figure 3e,f). After the administration of the *Bifidobacterium* species, the gut microbiota structure showed a transition towards the microbiota structure of the mice in the control group. However, this regulatory trend was not completely consistent with the ability of the *Bifidobacterium* species to alleviate the symptoms of T2D. Therefore, further analysis of the specific changes in the gut microbiota structure is required.

First, the effects of different intervention methods on the structure of the gut microbiota in the mice with T2D were analyzed at the phylum level (Figure 3g–m). The abundance of Firmicutes in the gut microbiota was significantly higher in the model group than in the control group, whereas the abundance of Bacteroidetes was lower in the model group. The F/B ratio and the abundance of Proteobacteria were also significantly higher in the model group than in the control group. Supplementation with *Bifidobacterium* strains increased the relative abundance of Actinobacteria and restored the abnormally elevated abundance of Proteobacteria to varying degrees in T2D mice. *B. adolescentis* 26M1, 30M5, and 2016 significantly increased the abundance of Actinobacteria, with strain 2016 showing the greatest effect. In summary, the gut microbiota of the mice in the model group was imbalanced at the phylum level, and the two *Bifidobacterium* species had different alleviating effects on this imbalance. However, not all strains that alleviated the symptoms of T2D significantly modulated the changes in Firmicutes and Bacteroidetes abundance or the F/B ratio induced by a high-fat diet and STZ.

Second, differential analysis of the gut microbiota at the genus level was performed using linear discriminant analysis effect size and random forest methods. As shown in (Figure 4a–d and Appendix A), *Anaerostipes*, *Coprococcus*, *Ruminococcus,* and *Bifidobacterium*, which are generally considered to be SFCA-producing bacteria [17], were the marker genera for different *B. adolescentis* interventions. Moreover, *B. bifidum* strains 9M10 and 26M7, which exhibited the ability to alleviate T2D symptoms in mice, also significantly regulated the abundance of these genera. The significant effect of these strains at alleviating the symptoms of T2D may be related to the increased abundance of SFCA-producing flora. In addition, the genus *Dysgonomonas*, which is associated with intestinal inflammation, was a marker genus of *B. adolescentis* 30M5 treatment. The abundance of *Dysgonomonas* has been shown to be positively correlated with obesity and nonalcoholic fatty liver disease [18].

Third, a combined analysis using the PICRUSt tool and the Kyoto Encyclopedia of Genes and Genomes (KEGG) database was used to better understand the changes in the metabolic function of the gut microbiota induced by the *B. adolescentis* strains. As shown in (Figure 4e), *B. adolescentis* 3M10 administration was predicted to down-regulate ether lipid metabolism and the retinoic acid-inducible gene I (RIG-I)-like receptor signaling pathway. Previous studies have shown that the down-regulation of RIG-I expression leads to the inhibition of downstream signaling pathways, including the p38 mitogen-activated protein kinase and nuclear factor kappa B (NF-KB) pathways, which ultimately leads to decreased production of a series of pro-inflammatory factors [19]. Therefore, *B. adolescentis* 3M10 may reduce low-grade inflammation by down-regulating the RIG-I-like receptor signaling pathway, thereby playing a role in relieving the symptoms of T2D. *B. adolescentis* 8M4 administration was predicted to significantly up-regulate the ubiquitin system, steroid hormone biosynthesis, steroid biosynthesis, and protein digestion and absorption pathways and down-regulate the flavone and flavonol biosynthesis and D-arginine and D-ornithine metabolism pathways (Figure 4f). The ubiquitin system is involved in multiple aspects of innate and adaptive immune responses, such as the regulation of inflammation and antigen receptor signaling [20]. Steroid hormone biosynthesis may affect the initiation of immune responses and may regulate immune cell function and inflammatory status [21]. The predicted results were similar for *B. adolescentis* 26M1, 41M3, 34M4, and 2016 and *B. bifidum* 26M7 (Appendix A). Other *B*. *bifidum* strains showed no particular functional trend. Overall, although these effective *Bifidobacterium* strains affected different pathways due to the changes they induced in the gut microbiota, they were all associated with effects on inflammation. Thus, although the specific effects of these strains on the gut microbiome were not identical and the pathways affected by the altered gut microbiome were not completely consistent, they may ultimately have a similar effect on inflammation.

### 3.4. Bifidobacterium Strains Alleviates the Symptoms of T2D via the Gut Microbiota-SCFA-Inflammation Axis

Low-grade inflammation is a common feature of patients with T2D, and there is much evidence pointing to a role of the immune system in the pathogenesis of T2D [22]. The predicted results of the alterations in the gut microbiota described in Section 3.3 also showed that the effective strains may relieve the symptoms of diabetes by regulating inflammation. An examination of serum inflammatory factors in mice revealed that a high-fat diet combined with STZ treatment induced inflammation, while supplementation with *B. adolescentis* strains decreased serum IL-6 (41M3 and 2016), TNF-α (3M10), or IFN-γ (26M1) concentrations and significantly increased the IL-10 (2016) concentration. *B. bifidum* strains 9M10 and 26M7 also decreased the serum IL-6 concentration in the mice with T2D (Figure 5a–d). In addition, the serum concentrations of proinflammatory factors TNF-α, IL-6 and IFN-γ were significantly positively correlated with blood glucose concentration and HOMA-IR values and were weakly positively correlated with liver index values and pancreatic pathological scores (Figure 5e). These findings indicate that the effect of *Bifidobacterium* strains on improving the disorders of blood glucose metabolism and insulin resistance may be related to the alleviation of inflammatory symptoms.

In addition to inflammation, the microbiome analysis results also showed that strains that were effective at alleviating T2D symptoms in mice were also able to up-regulate the abundance of SFCA-producing genera. SCFAs participate in the regulation of energy homeostasis through the G-protein coupled receptor (GPR)41/43 pathway and have the potential to prevent or alleviate metabolic disorders, such as obesity and T2D [23]. As shown in (Figure 5f–i), the concentrations of several SCFAs were lower in the model group than the control group. The fecal acetic acid and butyric acid concentrations significantly increased after supplementation with *B. adolescentis* (3M10, 26M1, and 41M3) or *B. bifidum* 9M10. Moreover, the increase in SCFA concentration by these strains was consistent with their effect on decreasing blood glucose concentration. The concentrations of propionic acid and butyric acid were negatively correlated with blood glucose indicators and insulin resistance, and negatively correlated with serum IL-6 and TNF-α concentrations (Figure 5j). The acetic acid concentration was weakly negatively correlated with blood glucose indicators and insulin resistance, and negatively correlated with serum IL-6, TNF-α, and IFN-γ concentrations. These findings indicate that the effects of *Bifidobacterium* species on hypoglycaemia and inflammation are related to their ability to increase the concentrations of acetic acid, propionic acid, and butyric acid.

### 3.5. The Greater Number and Stability of Core Genes, and a Unique Blood Sugar Regulation Gene May Give B. adolescentis an Advantage

To explore the differences in the ability of the two *Bifidobacterium* species to alleviate T2D symptoms in mice, we compared the genomes of 196 strains of *B. adolescentis* and *B. bifidum*, including 90 NCBI-derived genomes and the genomes of the eight strains used in the animal experiments in this study (Appendix A). The average number of genes in the two Bifidobacterium genomes is similar (*B. adolescentis*: *B. bifidum*, *1767*: *1736*), but the *B. adolescentis* genomes exhibited a higher proportion of homologous genes (61.06%: 54.67%), and the number was 13.70% (1079: 949) higher than that of *B. bifidum* (Figure 6a,b). In addition to the genes clustered into 663 core orthogroups, there were 240 and 180 unique core genes in *B. adolescentis* and *B. bifidum*, respectively (Figure 6c). Moreover, according to the pan-genome analysis results, when the number of genomes reached approximately 25, the number of *B. adolescentis* core genes no longer fluctuated greatly, and the number of pan-genome genes in the 98 strains reached 6707. *B. bifidum* showed a greater amount of fluctuation in the number of core genes, but had a smaller number of pan-genome genes (only 5697) (Figure 6d,e). This indicates that *B. adolescentis* may have acquired more functional genes and stably maintained a greater number of core genes during evolution.

Clusters of Orthologous Genes and KEGG databases were used to annotate and cluster the core genes of these two *Bifidobacterium* species. As there appeared to be no significant difference between the two species (Appendix A), then we conducted KEGG pathway enrichment analysis. The results showed a large number of *B. bifidum* core genes enriched in exosome and ribosome pathways, whereas genes enriched in transfer RNA biogenesis and exosome pathways had more significant functions in *B. adolescentis* (Figure 6f,g). For a more detailed exploration, we analyzed the unique regions (240 and 180 unique genes) of the genomes of these two *Bifidobacterium* species (Figure 6h–j). In addition to the organic systems class, *B. adolescentis* had more genes clustered in other KEGG pathway classes, especially the genetic information processing and human diseases classes. A greater number of genetic information processing genes (*B. adolescentis*: *B. bifidum*, 20:3) may be one of the reasons for the more stable genome of *B. adolescentis*. In the human diseases class, we found that the unique core genes of *B. adolescentis* were involved in the vancomycin resistance, cationic antimicrobial peptide resistance and antifolate resistance pathways, which may make this species more competitive in the gut environment. In addition, compared with *B. bifidum*, *B. adolescentis* had more genes involved in the two-component system, quorum sensing, biofilm formation and various carbon source and amino acid metabolism pathways, which appeared to confirm this competitive hypothesis (Appendix A). Most importantly, a unique *B. adolescentis* core gene (K00688, *glgP*, glycogen phosphorylase (EC:2.4.1.1)), which was annotated in the insulin signaling, insulin resistance, and glucagon signaling pathways, may explain the involvement of *B. adolescentis* in the decomposition of glycogen and the regulation of glycogen and the blood glucose balance. It is worth noting that this gene is a core gene shared by the 98 strains of *B. adolescentis* but is not present in *B. bifidum*. Furthermore, we have attempted to generate multiple *glgP* gene knockouts, but have not been able to obtain stable mutants due to a lack of technical support for *Bifidobacterium* gene manipulation (data not shown). Therefore, although the presence of this gene is a potential reason why *B. adolescentis* alleviates the symptoms of T2D in mice, it requires more verification.

## 4. Discussion

In recent years, the incidence of T2D and the associated mortality has been increasing due to unhealthy lifestyle factors, such as a lack of exercise and a high-fat, high-sugar diet. Animal and clinical studies have shown that the abundance of *Bifidobacterium* species is decreased in the intestines of individuals with T2D, while the intake of *Bifidobacterium* may relieve the symptoms of T2D. Here, we selected 16 strains of the two dominant *Bifidobacterium* species in the human intestines at different life stages, and used mice treated with a high-fat diet combined with STZ to determine whether *Bifidobacterium* has potential for use in the treatment of diabetes. Meanwhile, the inter-specific differences in the effects of two *Bifidobacterium* species at alleviating diabetes-related indicators were analyzed.

A large number of animal and clinical studies have shown abnormalities in the intestinal flora of diabetic mice or patients [24,25], such as an increase in the F/B ratio. In this study, a high-fat diet combined with STZ led to the disruption of gut microbiota homeostasis and a decrease in the microbiota species richness in diabetic mice, whereas most *Bifidobacterium* strains reversed this trend. The predicted microbiota functions suggest that the effective strains may play a role in relieving the symptoms of T2D by regulating inflammation. In fact, T2D is often accompanied by persistent chronic low-grade inflammation and there is much evidence indicating that the immune system plays a role in the pathogenesis of T2D [22]. Pro-inflammatory cytokines, such as TNF-α and IL-6, disrupt the insulin signaling pathway, and cause insulin resistance. The anti-inflammatory factor, IL-10, inhibits these pro-inflammatory effects and plays a role in protecting islet cells and insulin function [26,27]. Inhibiting the production of pro-inflammatory factors, such as TNF-α and IL-6, may improve insulin sensitivity and glucose homeostasis [28]. In this study, the strains that showed a strong hypoglycaemic effect also significantly decreased the concentrations of TNF-α, IL-6, and IFN-γ to varying degrees in mice with T2D. Moreover, the concentrations of serum proinflammatory factors in mice were significantly positively correlated with blood glucose concentration and HOMA-IR values, and weakly positively correlated with liver index values and pancreatic pathological scores. These findings indicate that the effect of *Bifidobacterium* strains in relieving the symptoms of T2D may be related to the alleviation of inflammatory symptoms.

The ability of *Bifidobacterium* strains to modulate inflammation in the T2D mice appeared to be determined by their ability to modulate host intestinal SCFAs. Intestinal SCFAs are mainly produced by the fermentation of undigested carbohydrates by gut microbes [29]. Previous studies have shown that probiotics with significant hypoglycaemic effects increase SCFA levels in mouse feces [30,31,32]. Acetic acid, one of the major SCFAs, has been shown to control appetite and inhibit body fat accumulation [33]. A high-fat diet supplemented with butyric acid prevents insulin resistance and obesity in C57BL/6J mice [34], and butyric acid has been shown to decrease the production of the pro-inflammatory cytokine and inhibit the activation of NF-κB signaling pathway [35]. In addition, propionic acid and butyric acid induce gluconeogenesis in the intestines to regulate energy production and maintain glucose homeostasis [36]. We found that supplementation with *B. adolescentis* 3M10, 26M1, or 41M3 or *B. bifidum* 9M10 or 26M7 significantly increased the concentrations of acetic acid and butyric acid. The increase in SCFA concentrations induced by these strains was consistent with their effects on blood glucose concentration, and SCFA concentration was negatively correlated with serum IL-6, TNF-α, and IFN-γ concentrations. Thus, *Bifidobacterium* species may increase the concentration of SCFAs, thus alleviating the inflammatory state of mice with T2D and further alleviating diabetic symptoms [37,38]. The abundance of SCFAs in the intestines is closely related to the structure of the gut microbiota. In this study, the *Bifidobacterium* strains that were effective at relieving the symptoms of T2D also increased the abundance of SCFA-producing bacteria in mouse feces. Specifically, *B. adolescentis* significantly increased the abundance of *Bifidobacterium* [39] (26M1, 30M5 and 2016), *Coprococcus* [40] (8M4 and 34M4), *Dorea* [41] (41M3), *Ruminococcus* [42,43] (26M1), and *Allobaculum* (3M10). *B. bifidum* 26M7 significantly increased the abundance of *Coprococcus*, *Roseburia,* and *Ruminococcus*, and *B. bifidum* 9M10 increased the abundance of *Coprococcus* and *Dorea* to a certain extent. The ability of these strains to regulate the microbiota was consistent with the actual SCFA concentrations in the samples. This also explains the advantages of B. adolescentis 26M1, 41M3, and 2016 at regulating blood glucose metabolism disorders, alleviating insulin resistance, and decreasing inflammation. Thus, *Bifidobacterium* strains found to be effective at relieving the symptoms of T2D may do so through the gut microbiota-SCFA-inflammatory axis.

Overall, a greater number of *B. adolescentis* strains showed a stronger regulatory effect on T2D in the present study. However, it is unclear why these two *Bifidobacterium* species had different abilities to modulate inflammation and intestinal SCFA production in mice with T2D. An analysis of 198 genomes of these two strains showed that *B. adolescentis* has a greater number of core genes and these genes are more evolutionarily stable. These core genes may enable *B. adolescentis* to acquire greater viability than *B. bifidum* in the gut of the host (especially adult hosts) through increased bacteriocin secretion, antibiotic resistance, the utilisation of multiple carbon sources, amino acid metabolism, quorum sensing, and biofilm formation [9,10]. This increase in viability may enable *B. adolescentis* to more readily occupy a dominant position in the gut and may make it more capable of restoring the homeostasis of the gut microbiota and increasing the abundance of SCFA-producing bacteria in mice with T2D. Furthermore, all *B. adolescentis* genomes contained the *glgp* gene, which encodes a glycogen phosphorylase that participates in the degradation of glycogen and the regulation of the host’s blood glucose balance [44]. However, whether this gene plays a key role in *B. adolescentis* alleviating T2D symptoms in mice still needs to be fully verified.

## 5. Conclusions

We studied the effects of two *Bifidobacterium* species that are dominant in the human gut at different life stages on T2D. Overall, a greater number of *B. adolescentis* strains showed good potential for decreasing FBG concentration and alleviating insulin resistance. The strains that were effective at relieving the symptoms of T2D showed some commonalities. In general, those with a significant hypoglycaemic effect relieved inflammation in T2D mice to a greater extent than those without such an effect. This effect occurred through the *Bifidobacterium*-gut microbiota-SCFA-inflammation axis. Besides, the unique glycogen phosphorylase gene, *glgP*, in the core genome of *B. adolescentis* is a potential mechanism by which this species relieves the symptoms of T2D. This gene is a potential molecular target for future studies of the mechanism of action of probiotics in the treatment of T2D. These findings provide ideas for future studies to identify probiotics for the treatment of metabolic diseases.

## Figures and Tables

**Figure 1 nutrients-14-02479-f001:**
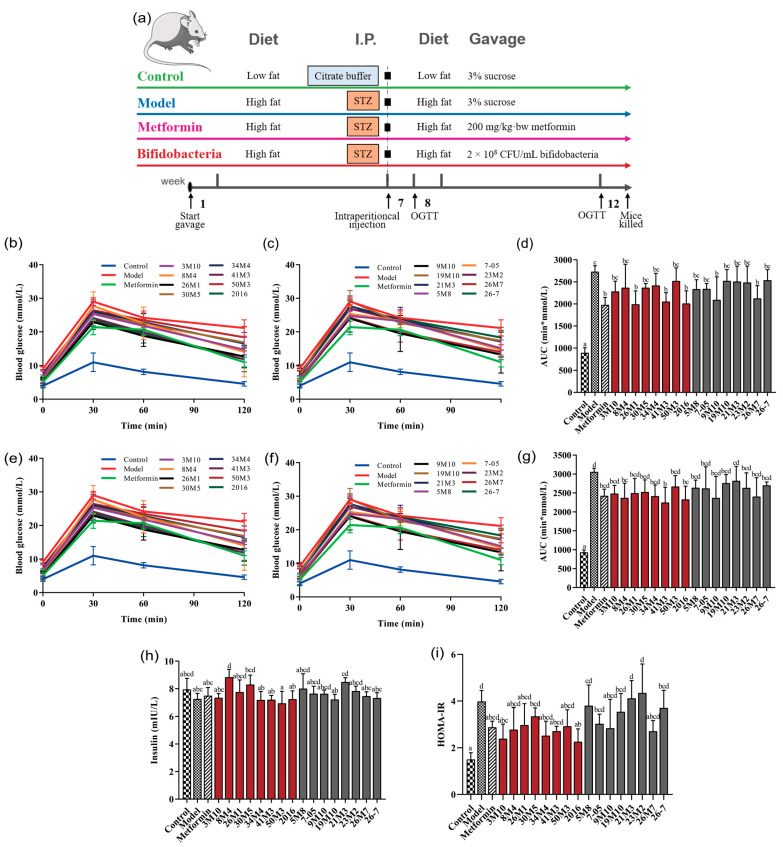
Effects of different *Bifidobacterium* on the regulation of glucose metabolism in T2D mice. (**a**) Animal experimental schedule; (**b**–**g**) OGTT and AUC_glucose_ at 1 or 5 weeks after STZ injection, respectively; (**h**) Insulin level; (**i**) HOMA-IR. Different letters represent statistically significant differences between different groups (*p* < 0.05), *n* = 6.

**Figure 2 nutrients-14-02479-f002:**
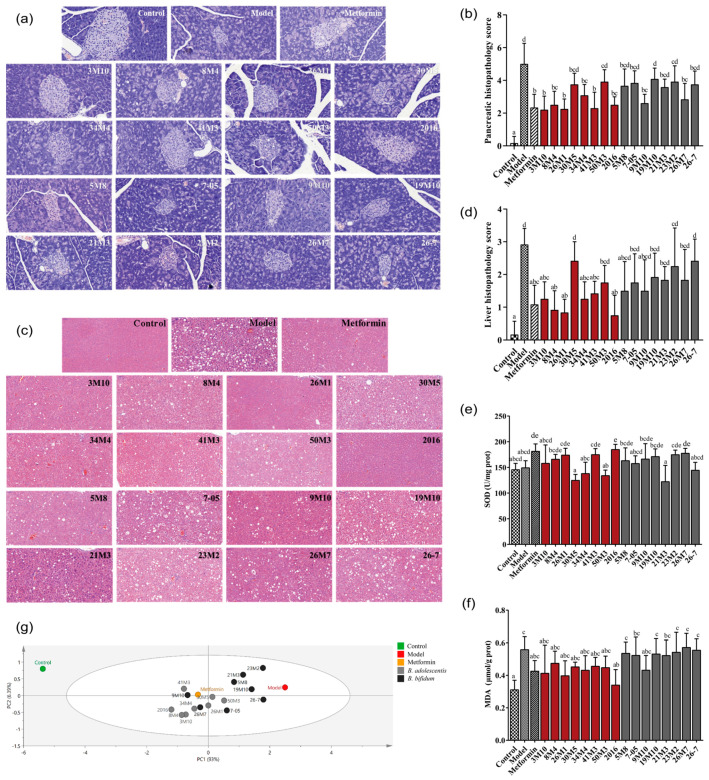
Effects of different *Bifidobacterium* on the pancreas and liver of T2D mice. (**a**) H&E staining of pancreas (400×); (**b**) Histopathological scores of pancreas; (**c**) H&E staining of the liver (200×); (**d**) Histopathological scores of liver; (**e**) liver SOD level; (**f**) liver MDA level; (**g**) Principal component analysis of blood glucose and lipid metabolism related index and pancreas and liver pathology. Different letters represent statistically significant differences between different groups (*p* < 0.05), *n* = 6.

**Figure 3 nutrients-14-02479-f003:**
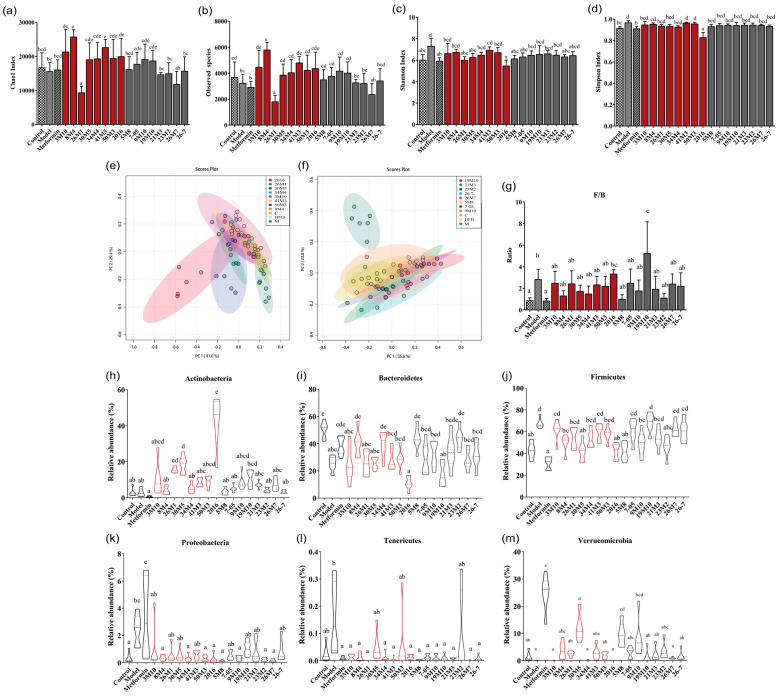
*Bifidobacterium* treatment altered gut microbiota diversity and structure. (**a**–**d**) Alpha diversity index; (**e**,**f**) PCoA plot based on beta diversity; (**g**–**m**) Relative abundance of different phylum. Different letters represent statistically significant differences between different groups (*p* < 0.05), *n* = 6.

**Figure 4 nutrients-14-02479-f004:**
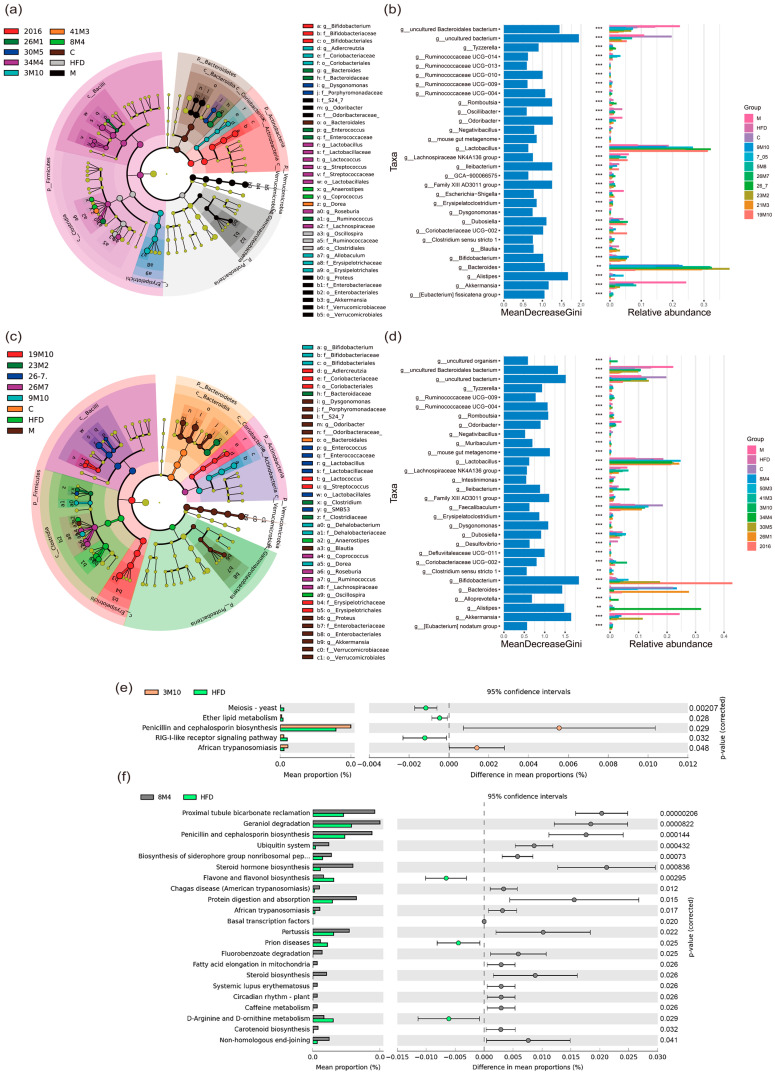
*Bifidobacterium* treatment altered gut microbiota at the family and genus levels. (**a**,**c**) LEfSe analysis used to label biomarkers in all sample groups; (**b**,**d**) Random forest analysis of genus level differences caused by different treatments; (**e**,**f**) Prediction of gut microbiota function. C = Control, HFD = Model, M = Metformin, ** *p* < 0.01, *** *p* < 0.001 in the unpaired *t*-tests.

**Figure 5 nutrients-14-02479-f005:**
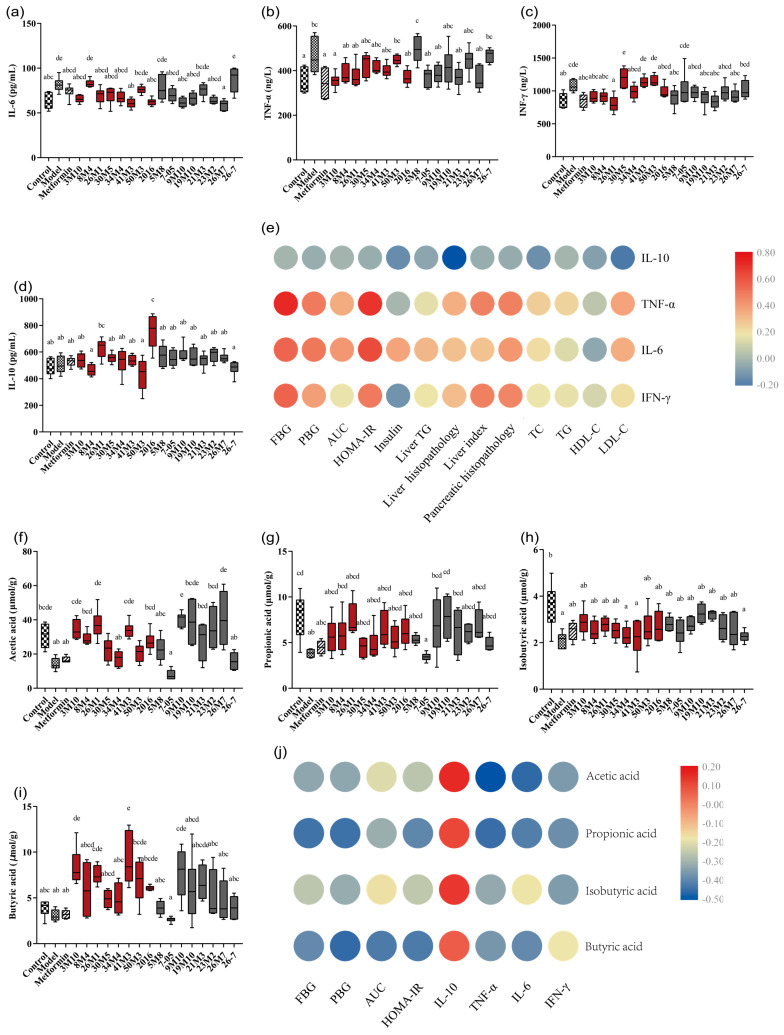
Effects of different *Bifidobacterium* on the inflammation and SCFAs regulation of T2D mice. (**a**–**d**) IL-6, TNF-α, INF-γ and IL-10 levels in serum; (**e**) Correlation analysis between inflammatory factors and blood glucose and lipid metabolism and visceral indexes; (**f**–**i**) Acetate, propitiate, isobutyrate, and butyrate levels in the feces; (**j**) Correlation analysis between SCFAs and blood glucose metabolism index and inflammatory factors. Different letters represent statistically significant differences between different groups (*p* < 0.05), *n* = 6.

**Figure 6 nutrients-14-02479-f006:**
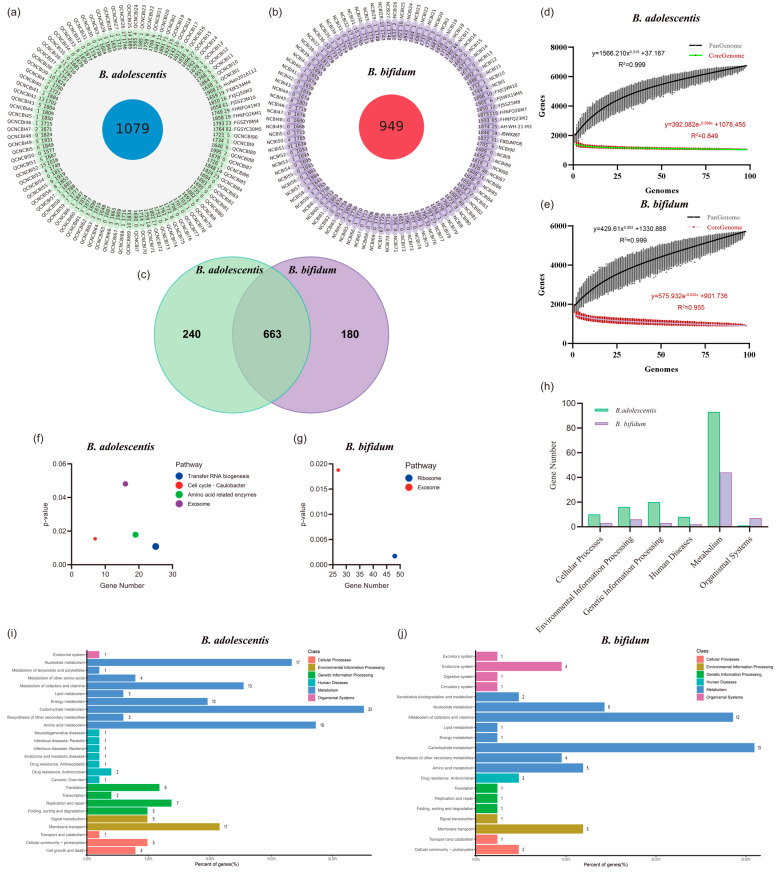
Genomic differences between two *Bifidobacterium*. (**a**,**b**) Homologous gene analysis; (**c**) Veen map of core genes; (**d**,**e**) Pan genomic analysis; (**f**,**g**) KEGG pathway enrichment analysis; (**h**–**j**) KEGG pathway cluster analysis of unique core genes.

**Table 1 nutrients-14-02479-t001:** Strain information used in this study.

Species	Strain	Number	Source	Culture Conditions
*B. adolescentis*	FJSSZ3M10	3M10	Human faeces	37 °C, MRS, anaerobic
*B. adolescentis*	FGSZY8M4 (CCFM1108)	8M4	Human faeces	37 °C, MRS, anaerobic
*B. adolescentis*	FHNFQ26M1	26M1	Human faeces	37 °C, MRS, anaerobic
*B. adolescentis*	FGSYC30M5	30M5	Human faeces	37 °C, MRS, anaerobic
*B. adolescentis*	FXJKS34M4	34M4	Human faeces	37 °C, MRS, anaerobic
*B. adolescentis*	FHNFQ41M3	41M3	Human faeces	37 °C, MRS, anaerobic
*B. adolescentis*	FXJCJ50M3	50M3	Human faeces	37 °C, MRS, anaerobic
*B. adolescentis*	HuNan112 (CCFM1261)	2016	Human faeces	37 °C, MRS, anaerobic
*B. bifidum*	FJSSZ5M8	5M8	Human faeces	37 °C, MRS, anaerobic
*B. bifidum*	FSDJN705	7-05	Human faeces	37 °C, MRS, anaerobic
*B. bifidum*	FXJCJ9M10	9M10	Human faeces	37 °C, MRS, anaerobic
*B. bifidum*	JSWX19M5	19M10	Human faeces	37 °C, MRS, anaerobic
*B. bifidum*	AHWH21M3	21M3	Human faeces	37 °C, MRS, anaerobic
*B. bifidum*	FHNFQ23M2	23M2	Human faeces	37 °C, MRS, anaerobic
*B. bifidum*	FHNFQ26M7 (CCFM1165)	26M7	Human faeces	37 °C, MRS, anaerobic
*B. bifidum*	JSWX267	26-7	Human faeces	37 °C, MRS, anaerobic

## Data Availability

The data sets generated during and/or analyzed during the current study are either shown in the manuscript and Appendix A or available from the corresponding author on reasonable request.

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
