# Peer review of "Bifidobacterium adolescentis Is Effective in Relieving Type 2 Diabetes and May Be Related to Its Dominant Core Genome and Gut Microbiota Modulation Capacity"

_nutrients, 2022, doi:10.3390/nu14122479_

Round 1

Reviewer 1 Report

Manuscript ID: nutrients-1765916

Bifidobacterium adolescentis is effective in relieving type 2 diabetes due to a dominant core genome and gut microbiota modulation capacity.

Authors: Xin Qian, Qian Si, Guopeng Lin, Minmin Zhu *, Jingyu Lu, Hao Zhang, Gang Wang and Wei Chen

General comment to the authors

This is an interesting manuscript, well conceived, well structured and the results are consistent with the conclusions. However, some aspects should be improved, especially those related to the figures that have low resolution and most of them are not even legible.

Specific Comments

- Introduction:

Line 43: Correct the sentence “The pathogenesis of T2D mainly manifests as islet function defects and insulin resistance caused by autoimmune destruction or insufficient sensitivity, which, in turn, leads to obvious hyperglycaemia” Type 2 diabetes develops as a consequence of insulin resistance rather than autoimmune islet destruction.

- Materials and methods:

   Preparation of bacterial strains:

Line 89: “change 2x108” to “2x108

   Animal experiments:

   This section is unclear. In Figure 1a there is the abbreviation OGTT (without specifying what it is in the caption, although it is obvious it should be) which is not mentioned in this section. On the other hand, the authors speak of PBG without speaking of OGTT, i.e. the reader reading the body of the manuscript have not clear what exactly they mean by PBG. In addition, I believe that this section should indicate the vehicle and the dose of STZ used. In addition, it is also necessary to mention in this section the approval of the ethics committee that there is at the end of the manuscript. In short, I think that the supplementary part is unnecessary if it is explained here concisely but explicitly.

   Finally, why did the authors use a dose of 100mg/kg bw when most of the literature describes 50mg/kg bw?

   Biochemical analysis:

   How glucose and insulin are measured is not described.

   Histopathological analysis:

   It would be more correct to put the magnification scale and not the lenses of the objective. In this section, you should indicate how many sections were analyzed to make the statistics shown in Figure 2b and c. was one section per animal or more analyzed? In addition, what was quantified to determine histopathology scores? Any details?

   SCFA analysis:

   The full name of the abbreviation SCFA I believe does not appear earlier in the text. It should be indicated.

- Results:

   In general, when there are no significant differences, there is no need to mention trends. There is simply no difference!

   Line 214-2016: The authors comment that strains 8M4, 26M1 and 2016 decrease liver weight with respect to the model. No significant differences with respect to the model are observed in table S2.

- Discussion:

   Line 440: "undigestible starch" change to "undigestible carbohydrates".

   Lines 445-446: NF-kB is not a cytokine!!!!

Figures.

   Figure 1: Overall, it is very crowded and it is difficult to see the results clearly.

First: In my opinion, the experimental design could be a separate figure, with a well-explained caption.

Second: I think it would improve if the FBG and PGB figures were removed and only the OGTT and AUC figures were shown. The figures of the FGB and PBG could be in supplementary material.

Third: Personally I am not able to observe any difference between figures: b and g; c and h; d and i; and e and j.

Figure 2: Mainly the g-section you cannot see anything, it lacks definition, when zoomed in it is all blurry.

Figure 3, 4, 6 and S1, S2, S3: Similar to the previous comment, they are blurry and some parts are too small, it cannot see anything!

Figure 5: In this case, there is a significant disproportion. Sections a, b, c, d, f, g, h and i are too small (blurry and difficult to see) compared to e and j (although these also do not have a good resolution, they are legible because they are larger!).

Author Response

We appreciate deeply every effort made by editors and reviewers for this manuscript. All the comments are valuable and very helpful for revising and improving our paper. Accordingly, we have carefully revised this manuscript according to your comments. In addition, in order to better highlight the modified parts, we added continuous line numbers in the manuscript and marked in red in the revised manuscript and the responses to reviewers’ comments are listed as follows:

Response to Reviewer #1:

Comment 1: Line 43: Correct the sentence “The pathogenesis of T2D mainly manifests as islet function defects and insulin resistance caused by autoimmune destruction or insufficient sensitivity, which, in turn, leads to obvious hyperglycaemia” Type 2 diabetes develops as a consequence of insulin resistance rather than autoimmune islet destruction.

Response: Sincerely thank you for your suggestion. We removed the description of autoimmune islet destruction.

“The T2D mainly manifests as insulin resistance, which, in turn, leads to obvious hyperglycaemia.” (line33-line34)

Comment 2: Line 89: “change 2x108” to “2x108

Response: We appreciate the reviewer for careful review work. We have completed the modification in the corresponding location (line 89) according to your suggestion.

Comment 3: Animal experiments: This section is unclear. In Figure 1a there is the abbreviation OGTT (without specifying what it is in the caption, although it is obvious it should be) which is not mentioned in this section. On the other hand, the authors speak of PBG without speaking of OGTT, i.e. the reader reading the body of the manuscript have not clear what exactly they mean by PBG. In addition, I believe that this section should indicate the vehicle and the dose of STZ used. In addition, it is also necessary to mention in this section the approval of the ethics committee that there is at the end of the manuscript. In short, I think that the supplementary part is unnecessary if it is explained here concisely but explicitly. Finally, why did the authors use a dose of 100mg/kg bw when most of the literature describes 50mg/kg bw?

Response: We sincerely appreciate the reviewer for the constructive comment and positive comments on animal experiments of our work. We have reorganized the description of the animal experiment procedure on line 99-109. Specifically, First, postprandial blood glucose (PBG) refers to the blood glucose level at 2 h during the OGTT, and we deleted the relevant content after realizing that this description was ambiguous and unnecessary. Second, in the preliminary experiment, the success rate of establishing T2D model by 50mg/kg·bw intraperitoneal injection of STZ was only 40% (2/5). In the formal experiment, in order to improve the success rate of the T2D model establishment, the dosage of STZ was increased to 100mg/kg·bw. Third, we added the relevant ethics numbers at the end of this section.

“Mice in each group were fed a high-fat diet (the control group was fed a low-fat diet) and were administered metformin or a Bifidobacterium suspension by gavage for 12 consecu-tive weeks. The feed and gavage formulations are detailed in Supplementary Material 1.1. On the first day of week 7, the mice, except those in the control group, were injected with STZ (Sigma, St Louis, MO, USA). Mice in all groups were fasted for 12 hours in advance, the Control group were intraperitoneally injected with 50 mmol/L citric ac-id-sodium citrate buffer (pH 4.5), the other groups were intraperitoneally injected with STZ dissolved in 50 mmol/L citric acid-sodium citrate buffer at the dose of 100 mg/kg·bw. Then, blood glucose levels were determined at 0, 0.5, 1.0, and 2.0 h. The area under curve (AUC) was calculated as follows: AUC = 0.25 × (G0 h + G0.5 h) + 0.25 × (G0.5 h + G1.0 h) + 0.5 × (G1.0 h + G2.0 h). All of the animal procedures were performed in ac-cordance with the experimental animal management regulations of the Animal Welfare and Ethics Committee of Jiangnan University (JN.No20190615c1261115).” (line 99-109)

Comment 4: How glucose and insulin are measured is not described.

Response: Thank you for your professional suggestion. We have added a description of the methods for glucose and insulin assay in Section 2.4.

“Blood glucose was measured with Roche glucometer and blood glucose test strips by means of tail blood collection with a blood collection needle.” (line 121-123)

“The serum concentrations of insulin … … were measured using an enzyme-linked im-munosorbent assay kit (Senbeijia Biological Technology, Nanjing, China).” (line 126)

Comment 5: It would be more correct to put the magnification scale and not the lenses of the objective. In this section, you should indicate how many sections were analyzed to make the statistics shown in Figure 2b and c. was one section per animal or more analyzed? In addition, what was quantified to determine histopathology scores? Any details?

Response: Thank you for your professional suggestion. In Section 2.5, we modified the description of the magnification scale. The specific number of pathological sections and the quantitative indicators of histopathology scores were clarified.

“Stained pancreas and liver tissues were observed under a microscope with 400× and 200× magnification scale, respectively. All morphological lesions were rated on a scale of 1–5 by their severity according to the NTP Pathology guidelines. Tissue sections with no lesions were marked as 0. Pancreatic tissues were scored for exocrine and endocrine. The exocrine scores were mainly based on the morphology of pancreatic cells, and whether there were vacuolar deformation and turbidity. The endocrine scores were mainly based on the number and the shape of the islets, and whether the cells showed nuclear condensation. The final score was the sum of the exocrine and endocrine scores. The scores of liver tissue lesions were mainly based on the status of steatosis, including the size and density of fatty vacuoles in the liver tissue. The pancreatic tissue and liver tissue of each mouse were all scored in a blinded manner. (line 133-142)

Comment 6: The full name of the abbreviation SCFA I believe does not appear earlier in the text. It should be indicated.

Response: Thank you for careful review work. We added the full name of SCFAs in line 145.

“Short-chain fatty acids (SCFAs) were …” (line 144)

Comment 7: In general, when there are no significant differences, there is no need to mention trends. There is simply no difference!   Line 214-2016: The authors comment that strains 8M4, 26M1 and 2016 decrease liver weight with respect to the model. No significant differences with respect to the model are observed in table S2.

Response: Thank you for your professional suggestion. We are very sorry for the inadvertent confusion of “liver/body weight” with “liver weight”, which has now been corrected in line 223 and 228.

Comment 8: Line 440: "undigestible starch" change to "undigestible carbohydrates".

Response: Thank you for your professional suggestion. "undigestible starch" has beeb changed to "undigestible carbohydrates".

Comment 9: Lines 445-446: NF-kB is not a cytokine!!!!

Response: We thank the reviewer for your carefulness to point out our incorrect expressions that save us from awkward. We have corrected the incorrect content.

“… butyric acid has been shown to decrease the production of the pro-inflammatory cytokine and inhibit the activation of NF-κB signaling pathway.” (line 459)

Comment 10: Figure 1: Overall, it is very crowded and it is difficult to see the results clearly.

First: In my opinion, the experimental design could be a separate figure, with a well-explained caption.

Second: I think it would improve if the FBG and PGB figures were removed and only the OGTT and AUC figures were shown. The figures of the FGB and PBG could be in supplementary material.

Third: Personally I am not able to observe any difference between figures: b and g; c and h; d and i; and e and j.

Figure 2: Mainly the g-section you cannot see anything, it lacks definition, when zoomed in it is all blurry.

Figure 3, 4, 6 and S1, S2, S3: Similar to the previous comment, they are blurry and some parts are too small, it cannot see anything!

Figure 5: In this case, there is a significant disproportion. Sections a, b, c, d, f, g, h and i are too small (blurry and difficult to see) compared to e and j (although these also do not have a good resolution, they are legible because they are larger!).

Response: Sincerely thank you for your suggestion. We have rearranged Figure 1 and improved the size and clarity of all images in the manuscript as much as possible. In order to prevent the reduction of image resolution caused by submission system, we have also uploaded high resolution PDF versions of each figure (scalable vector illustration) in the attachment.

Reviewer 2 Report

Although the work is very interesting in some parts it is very difficult to read and interpret the figures. Simplify the figures, especially figure 1.

Author Response

We appreciate deeply every effort made by editors and reviewers for this manuscript. All the comments are valuable and very helpful for revising and improving our paper. Accordingly, we have carefully revised this manuscript according to your comments. In addition, in order to better highlight the modified parts, we added continuous line numbers in the manuscript and marked in red in the revised manuscript and the responses to reviewers’ comments are listed as follows:

Response to Reviewer #2:

Comment 1: Although the work is very interesting in some parts it is very difficult to read and interpret the figures. Simplify the figures, especially figure 1.

Response: Sincerely thank you for your suggestion. We have rearranged Figure 1 and improved the size and clarity of all images in the manuscript as much as possible. In order to prevent the reduction of image resolution caused by submission system, we have also uploaded high resolution PDF versions of each figure (scalable vector illustration) in the attachment.